# Technical Efficiency of Public and Private Hospitals in Beijing, China: A Comparative Study

**DOI:** 10.3390/ijerph17010082

**Published:** 2019-12-20

**Authors:** Rize Jing, Tingting Xu, Xiaozhen Lai, Elham Mahmoudi, Hai Fang

**Affiliations:** 1School of Public Health, Peking University, Beijing 100083, China; rzjing2015@hsc.pku.edu.cn (R.J.);; 2China Center for Health Development Studies, Peking University, Beijing 100083, China; 3Department of Social and Behavioral Science, Harvard TH Chan School of Public Health, Harvard University, 677 Huntington Avenue, Boston, MA 02115, USA; 4Department of Family Medicine, University of Michigan, Ann Arbor, MI 48109, USA; 5Peking University Health Science Center-Chinese Center for Disease Control and Prevention Joint Center for Vaccine Economics, Peking University, Beijing 100083, China; 6Key Laboratory of Reproductive Health, National Health Commission of the People’s Republic of China, Peking University, Beijing 10083, China

**Keywords:** technical efficiency, data envelopment analysis, public hospital, private hospital, China

## Abstract

*Objective:* With the participation of private hospitals in the health system, improving hospital efficiency becomes more important. This study aimed to evaluate the technical efficiency of public and private hospitals in Beijing, China, and analyze the influencing factors of hospitals’ technical efficiency, and thus provide policy implications to improve the efficiency of public and private hospitals. *Method:* This study used a data set of 154–232 hospitals from “Beijing’s Health and Family Planning Statistical Yearbooks” in 2012–2017. The data envelopment analysis (DEA) model was employed to measure technical efficiency. The propensity score matching (PSM) method was used for matching “post-randomization” to directly compare the efficiency of public and private hospitals, and the Tobit regression was conducted to analyze the influencing factors of technical efficiency in public and private hospitals. *Results:* The technical efficiency, pure technical efficiency and scale efficiency of public hospitals were higher than those of private hospitals during 2012–2017. After matching propensity scores, although the scale efficiency of public hospitals remained higher than that of their private counterparts, the pure technical efficiency of public hospitals was lower than that of private hospitals. Panel Tobit regression indicated that many hospital characteristics such as service type, level, and governance body affected public hospitals’ efficiency, while only the geographical location had an impact on private hospitals’ efficiency. For public hospitals in Beijing, those with lower average outpatient and inpatient costs per capita had better performance in technical efficiency, and bed occupancy rate, annual visits per doctor, and the ratio of doctors to nurses also showed a positive sign with technical efficiency. For private hospitals, the average length of stay was negatively associated with technical efficiency, but the bed occupancy rate, annual visits per doctor, and average outpatient cost were positively associated with technical efficiency. *Conclusions:* To improve technical efficiency, public hospitals should focus on improving the management standards, including the rational structure of doctors and nurses as well as appropriate reduction of hospitalization expenses. Private hospitals should expand their scale with proper restructuring, mergers, and acquisitions, and pay special attention to shortening the average length of stay and increasing the bed occupancy rate.

## 1. Introduction

The ever-increasing health expenditure is an important health policy problem in China and across the world. Hospital costs contribute substantially to increasing health spending. In 2015, hospital costs accounted for nearly 40% of overall health expenditure in OECD countries [1], while that of China reached up to 63% [2]. Controlling hospital costs is considered as a key issue, and previous studies have shown that the uncontrollable growth in health expenditure mainly resulted from demographic change and technological progress, both of which are difficult to manage for policy-makers. The third factor contributing to high costs was inefficiency in health care delivery and could be controlled [3]. Therefore, improving hospital efficiency is the main target of hospital managers and policy-makers.

As the reform and opening-up policy began in 1978, private hospitals have shown remarkable progress as more social resources flow into the medical industry in China [4]. China has launched a nationwide healthcare reform in 2009, in which the government encouraged private investment in the healthcare sector [5]. The motivation behind privatization can be interpreted as a strategic move to improve the performance of public hospitals via competing against the private sector [6]. The public hospitals are funded by governments, such as the National Health Commission of China, provincial, municipal, county or district governments, and public enterprise or institution, while the private hospitals are funded by private enterprises or individuals. Although private hospitals have existed for a long time in China, they still played a minor role in the health system and health care provision. However, some companion policies putting private hospitals on equal footing with public hospitals have been introduced. First, private hospitals can charge the same co-payment rates with public hospitals if they have signed contracts with health insurance programs. Therefore, both public and private hospitals are paid by health insurance and patients’ out-of-pocket money through fee-for-service [7], and patients can choose either of them at their wish. Second, physicians working in private hospitals embraced the opportunity to promote within the medical professional ranking system [5], and physicians in public hospitals with a multi-sited license could work in private hospitals. Third, a faster approval process and tax exemptions encouraged the entry of non-for-profit private hospitals into the markets [5,8]. At the same time, there were some extra constraints for public hospitals such as the banning of new building-construction projects or the expansion of beds. From 2009 to 2017 in China, the number of private hospitals increased from 6240 (30.75% of the total number of hospitals) to 18,759 (60.40%) [9,10], the market share for outpatient services in the private sector grew from 8% to 14.2%, and the market share for inpatient services in the private sector increased from 8% to 17.6% [10]. Therefore, even if the scale of most private hospitals cannot compete with that of tertiary public hospitals [11], the greater participation of private hospitals in the health system can still help boost the vitality of the health market and promote the hospitals’ efficiency.

Farrell [12] introduced technical efficiency, which was related to a best-performance frontier determined by a representative peer group. Technical efficiency indicates the extent to which a hospital is minimizing inputs to achieve its chosen output level or maximizing outputs given a chosen input level [13]. The present study used technical efficiency to measure hospital efficiency. Only a few studies explored the hospital efficiency in China before the new healthcare reform in 2009, and their findings were still inconclusive and inconsistent [14,15]. Therefore, it is worthwhile to evaluate the technical efficiency of private and public hospitals after China’s healthcare reform in 2009.

This study focused on the difference in efficiency between public and private hospitals in Beijing, China. Beijing has rich medical resources in both public and private health sectors, and consequently, hospitals in Beijing not only provide medical services for local residents but also undertake the duty of treating intractable diseases around the country. After the new healthcare reform in 2009, Chinese private hospitals began to spring up, especially in Beijing. In 2012, there were 254 public hospitals and 339 private hospitals [16], and in the next years, private hospitals were still increasing. Most public hospitals in China have to earn 90% of their revenues from services provided, with direct subsidies from the government making up the rest, which is popular not only in Beijing, but across China [6]. In addition, since the announcement of promoting hospitals with social capitals in China, private hospitals have sprung up in all urban areas. Therefore, this study chose Beijing as a representative to compare the trend of efficiency in public and private hospitals, which can provide some information and inspiration to other urban parts of China, and even contribute to the development of public and private hospitals in other developing countries.

There are three objectives in the present study: (1) Comparing the technical efficiency, pure technical efficiency and scale efficiency between public and private hospitals in China. (2) Exploring the effects of hospital ownership on efficiency. (3) Analyzing influencing factors of public and private hospitals’ efficiency and proposing suggestions to improve hospital efficiencies.

## 2. Literature Review

There are many studies examining the impacts of hospital ownership on technical efficiency, in which data envelopment analysis (DEA) is the best-known and widely-adopted method. Hollingsworth found that public hospitals could provide medical services in a more efficient way than private ones in Europe and the United States by reviewing 317 published papers [17]. Shen et al. [18] confirmed the effects of hospital ownership on efficiency using meta-analytic methods, but they could not draw a conclusion that private hospitals operated more efficiently.

Ozcan et al. [19] and Burgess and Wilson [20] drew a consistent conclusion by using DEA that the efficiency of public hospitals was higher than that of private ones, and showed a difference in efficiency between private for-profit and non-profit hospitals. On the other hand, Brown [21] reported a higher efficiency of private hospitals than public hospitals. Some intrastate or interstate studies in the United States also showed mixed results. Ferrier et al. [22] used data from the southwest United States and revealed that private for-profit hospitals performed more efficiently than public hospitals, while Chirikos and Sear [23] drew the opposite conclusion using data from Florida state. However, other studies did not report any influence of ownership on hospital efficiency [24,25].

A study from Germany indicated that the efficiency of public hospitals increased by 2.9–4.9% after privatization [26], while Helimig et al. [27] found higher efficiency in public and welfare hospitals compared with private hospitals. After comparing 128 public and private non-profit hospitals by DEA, an Austria study confirmed a significant link between technical efficiency and ownership types with multiple regression analysis and found that private non-profit hospitals were more efficient than the public [28]. Therefore, empirical studies worldwide have not reached a consistent conclusion on the efficiency of hospitals with different ownerships.

With few exceptions, most published papers did not examine the changing trend of efficiency under the competition between the private and public sectors. A few studies investigated how hospital ownership and other aspects of hospital market composition affect health care productivity, and found that for-profit hospitals had important spillover benefits for medical productivity [29,30]. Some other studies showed that private for-profit hospitals were more efficient in management, leading to improvements in management efficiency gains [31,32]. However, previous studies on hospital efficiency in China mainly focused on public hospitals [33,34,35] due to the slow development of private hospitals before 2009. There were only a few studies in China exploring the impact of hospital ownership on efficiency. Chen found that in Taiwan, China, private hospitals performed more efficiently than public hospitals, but in mainland China, public hospitals had higher efficiency compared to private hospitals [36]. Several studies speculated that for-profit private hospitals entering the market would compete with established public hospitals, which drove public hospitals to change their behavior and improve efficiency by offering higher physician salaries and acquiring the latest high-tech equipment [6]. However, there has been no empirical study proving the efficiency changes of public and private hospitals as private hospitals have been developing rapidly.

## 3. Materials and Methods

### 3.1. Data

Using a data set of public and private hospitals in 2012–2017 from “Beijing’s Health and Family Planning Statistical Yearbooks,” this study included secondary and tertiary hospitals in Beijing, capital of China. The total number of hospitals has increased from 154 in 2012 to 232 in 2017. All public hospitals are non-profit, while private hospitals can be for-profit or non-profit. Based on hospitals’ size and function, Chinese hospitals are classified into three levels, namely primary, secondary and tertiary levels. Secondary and tertiary hospitals undertake a large proportion of medical services, and thus they own 100–499 and over 500 ward beds, respectively. As Table 1 showed, there were 135 secondary and tertiary public hospitals in 2012, while the number of secondary and tertiary private hospitals was only 19. However, private hospitals quickly entered into the health system after 2012. Specifically, the number of secondary and tertiary private hospitals in Beijing grew rapidly from 19 in 2012 to 87 in 2017, while the number of their public counterparts increased slowly from 135 in 2012 to 145 in 2017. In the present study, public and private primary hospitals were excluded to ensure the homogeneity of hospitals’ characteristics. The data set showed the service type of hospital (general hospitals, specialized hospitals, and traditional Chinese hospitals) and the geographical location (urban hospitals and suburban hospitals), and also included the number of beds and medical staff, outpatient/emergency visits, inpatient discharges, hospital revenues, hospital expenditures, and productivity indicators (bed occupancy rate, average length of stay, and annual visits per physician).

### 3.2. Data Envelopment Analysis

Data envelopment analysis (DEA) is a nonparametric frontier method that uses linear programming to construct a frontier of observed data and evaluates the relative technical efficiency of an individual hospital based on this frontier [37]. DEA allows for simultaneous consideration of multiple inputs and outputs, which is suitable for measuring the efficiency of complex service organizations like hospitals. In 1978, Charnes, Cooper, and Rhodes put forward the DEA model and named it with the initials of their surnames as “CCR.” The CCR model assumes a constant return to scale (CRS), in which an increase in the input will result in a proportionate increase in the output(s) [38]. The calculated efficiency in this model is therefore referred to as technical efficiency. In this manner, the producers are able to scale the inputs and outputs linearly without increasing or decreasing efficiency. Nevertheless, as not every hospital operates on a constant return to scale, technical efficiency will be influenced by the scale of hospitals. Scale efficiency calculates the extent to which a hospital is operating at a scale that maximizes the ratio of outputs to inputs, and any larger or smaller scale would lead to decreasing or increasing returns to scale. However, when hospitals are not operating at the optimum scale, technical efficiency measured with the CCR model may be altered by scale efficiency. In contrast, the BCC model proposed by Banker, Charnes, and Cooper [39] assumes variable returns to scale (VRS) so that the calculation of technical efficiency is unaffected by scale. The calculated efficiency is called pure technical efficiency. The DEA model can be divided into input-oriented and output-oriented models according to different measurements of efficiency. The input-oriented model aims to reduce inputs, which is inconsistent with the current context of insufficient health resource investment in hospitals, so the output-oriented model is adopted in this study, in which scale efficiency equals technical efficiency divided by pure technical efficiency [40]. The efficiency score was calculated by MaxDEA software. The CCR model can be defined as follows:$$max\;\theta \left(\mu ,\vartheta \right)=\mu_{1}y_{1o}+\cdots +\mu_{s}y_{\mathrm{so}}.$$

Subject to:$$\vartheta_{1}x_{1o}+\cdots +\vartheta_{m}x_{mo}=1;\;\mu_{1}y_{1j}+\cdots +\mu_{s}y_{sj}\leq \vartheta_{1}x_{1j}+\cdots +\vartheta_{m}x_{mj},\;j=1,\ldots ,n,$$
$$\vartheta_{1},\;\vartheta_{2},\ldots ,\vartheta_{m}\geq 0;\mu_{1},\;\mu_{2},\ldots ,\mu_{s}\geq 0,$$
where θ is technical efficiency index, *x* represents input indicators, *y* represents output indicators, *v* represents input weight, and *μ* represents output weight.

The indicators of inputs and outputs were chosen in the light of the literature review. The supply of medical services requires the input of production factors, which were classified on the basis of Marshall’s basic classification criteria for production factors in this study [41]. We took the number of total health technicians, including the number of doctors, nurses, pharmacists, laboratorians, and other staff, as a labor factor, and adopted the number of beds as a capital factor. Some literature which reviewed the inputs and outputs of hospitals in measuring technical efficiency using DEA also found that the number of beds and medical staff were the most frequently used input indicators [42]. Three output indicators, including the number of outpatient visits, the number of inpatient discharges, and revenue, were determined according to Hu et al. [43], Tlotlego et al. [44], Cheng et al. [45], Zhang et al. [46], and Wang et al. [47]. The number of outpatient visits and inpatient discharges were regarded as the main outputs for hospitals with different types of ownership. Many hospitals in China, although in the name of being public, were in effect operating as private entities, putting profit above patient welfare, and thus their medical staff could be regarded as the residual claimants of profits [48,49]. Although maximizing revenues is inappropriate for public hospitals, many of them take it as an important measure for their performance in China. Therefore, the revenue is also enrolled as an output indicator in the present study. In addition, hospitals provide services to patients with different case-mix variation and quality, but there were no available data to accurately measure the case-mix indicators in our data set. The descriptive analysis of inputs and outputs was performed using STATA V.13.0 (Stata Corporation LLC, College Station, TX, USA).

### 3.3. Propensity Score Matching

Under the current healthcare system in China, the two ownership types of hospitals vary in scale, level, service type, number of beds, personnel, and so on, therefore it is difficult to distinguish the impact of ownership from other factors. The propensity score matching (PSM), proposed by Rosenbaum and Rubin [50], can create a “quasi-random” test. The propensity score is the conditional probability of assignment to a particular treatment given a vector of observed covariates. Adjusting for the scalar propensity score is sufficient to remove bias caused by all observed covariates, and Rosenbaum and Rubin [51] advised us to use the Logit model to estimate the propensity score. The Logit model in the present study included eight covariates: hospitals’ geographical location, level, service type, the number of beds, the number of health technicians, revenues, average outpatient cost, and inpatient cost per capita. The principle of “returning instead of replacing” was adopted due to the large differences between public and private hospitals. The equation of propensity score matching model was as follows:$$\mathrm{logit}\;\mathrm{private}=\beta_{0}+\beta_{1}\mathrm{urban}+\beta_{2}\mathrm{tertiary}+\beta_{3}\mathrm{type}+\beta_{4}\mathrm{beds}+\beta_{5}\mathrm{technicians}+\beta_{6}\mathrm{revenues}+\beta_{7}\mathrm{outpient}\;\mathrm{cost}+\beta_{8}\mathrm{inpatien}\;\mathrm{cost}+\theta_{i}.$$

The PSM was conducted using the STATA V.13.0 software.

### 3.4. Panel Tobit Regression Model with Random Effects

A panel Tobit regression model with random effects was applied to relate the technical efficiency to a number of explanatory variables. Censored efficiency scores (0–1, concentrate on boundary values) cannot be obtained via the ordinary least squares method, so it is preferable to estimate coefficients using the panel Tobit model. As there are insufficient statistics of individual heterogeneity allowing the fixed effects to be conditioned out of likelihood, the Tobit model with fixed effects is usually not recommended. A previous study developed a semiparametric estimator for unconditional fixed-effects Tobit model, showing that this model may be fit with the Tobit command with indicator variables for the panels, but the estimates are biased [52]. We divided influencing factors into two categories: hospital characteristics and internal management factors. Hospital characteristics [45,53] include service type (general, specialized, and traditional Chinese medicine), hospital-level (secondary and tertiary), governance body (for public hospitals), profitability (for private hospitals), and geographical location (urban and suburb). Moreover, six internal management factors include bed occupancy rate, the average length of stay, annual visits per doctor, and the ratio of doctors to nurses, average outpatient cost, and inpatient cost per capita [45,54]. The estimated Tobit model was as follows:(1)$$TE_{it}=\beta_{0}+\beta_{1}\mathrm{urban}+\beta_{2}\mathrm{tertiary}+\beta_{3}\mathrm{type}+\beta_{4}\mathrm{governance}\;\left(\mathrm{profit}\right)+\beta_{5}x_{it}+\varepsilon_{it},$$
where $TE_{it}$ is the annual technical efficiency, and other dummy variables are geographical location (0 = in urban area, 1 = in the suburb), hospital-level (0 = secondary, 1 = tertiary), service type (reference group: general hospitals), governance body (for public hospitals, reference group: commission hospitals), and profitability (for private hospitals, 0 = non-profit, 1 = for-profit). $X_{it}$ represents the internal management factors each year, and $\varepsilon_{it}$ is the error term. The panel Tobit regression model with random effects was conducted using STATA V.13.0 (Stata Corporation LLC, College Station, TX, USA).

## 4. Results

### 4.1. Descriptive Statistics

Table 1 showed the statistical description of inputs and outputs variables in public and private hospitals in Beijing from 2012 to 2017. First, the input and output indicators of public hospitals in Beijing were much higher than those of private hospitals in 2012–2017. Second, from 2012 to 2017, the average number of beds and total health technicians in public hospitals increased by 10.8% (from 529 to 586) and 12.4% (from 854 to 960), respectively, while those of private hospitals decreased by 23.5% (from 162 to 124) and 27.5% (from 247 to 179), respectively. Thirdly, the average number of outpatient and emergency visits, discharged patients, and revenue in public hospitals increased by 7.8% (from 798,551 to 861,042), 35.9% (from 14,209 to 19,310), and 11.3% (from 57,271 to 63,739), respectively, while the three indicators in private hospitals declined by 49.1% (from 138,163 to 70,339), 19.7% (from 2614 to 2099), and 35.7% (from 12,580 to 8095), respectively. The statistical description of explanatory variables could be found in Table A1. 

### 4.2. DEA Analysis

Table 2 showed the average technical efficiency, pure technical efficiency and scale efficiency scores of public and private hospitals in Beijing during 2012–2017. Using MaxDEA to analyze the data in 2012–2017, we found that public hospitals operated more efficiently than private hospitals in terms of all the three indicators, except in 2013, the pure technical efficiency of private hospitals (0.606) was higher than that of public hospitals (0.576). It helped reach the conclusion that public hospitals performed better in efficiency compared to private hospitals. The technical efficiency, pure technical efficiency, and scale efficiency of public hospitals had a downward trend, with technical efficiency decreasing from 0.589 in 2012 to 0.473 in 2017, pure technical efficiency from 0.612 to 0.518, and scale efficiency from 0.952 to 0.925. At the same time, technical efficiency and pure technical efficiency in private hospitals also declined from 0.452 and 0.606 in 2012 to 0.315 and 0.376 in 2017, respectively, but scale efficiency of private hospitals has risen from 0.782 in 2012 to 0.841 in 2017. In terms of different hospital levels, both pure technical and scale efficiency of tertiary public hospitals were higher than those of private hospitals, and for secondary public hospitals, the pure technical efficiency was lower than private hospitals, while the scale efficiency was better than that of private hospitals (Table A2 and Table A3 in Appendix A).

Table 3 indicates the number of different scales of return in public hospitals and private hospitals in Beijing from 2012 to 2017. A hospital that keeps constant returns to scale achieves the scale efficiency. Most private hospitals did not achieve scale efficiency for having increasing returns to scale. For example, among the 87 private hospitals in 2017, 80 were in the state of increasing returns to scale. Similarly, there were only a few hospitals that achieved scale efficiency, but the number of public hospitals with decreasing returns to scale surpassed that of hospitals with increasing returns to scale. We could also find in Table 2 that although the pure technical efficiency of private hospitals was generally lower than that of public hospitals, the gap was relatively small in some years. Therefore, it can be inferred that the main reason for the lower efficiency of private hospitals lies in their lower scale efficiency, and we should particularly focus on the scale of private hospitals in further discussion.

### 4.3. PSM Analysis

Table 4 showed the PSM results of the efficiency of private and public hospitals in 2012–2017. The huge differences between public and private hospitals in China under the current healthcare system have been widely acknowledged. Therefore, the PSM technology was further applied to compare the adjusted efficiency values of public and private hospitals. Through the post-randomization of data in 2012–2017 using a 1:4 case-control matching method, we found that before matching, the pure technical efficiency of public hospitals was higher than that of private hospitals from 2012 to 2017. After matching, however, public hospitals had lower pure technical efficiency than private ones every year. As for scale efficiency, public hospitals performed better than private ones before matching, and still showed better performance after matching (Table 4). After PSM, this study showed that the major cause of the inefficiency of private hospitals was their lower scale efficiency. Further analysis of the returns to scale revealed that most private hospitals that have not achieved scale efficiency were in the state of increasing returns to scale (Table 2). In addition, we used other matching methods such as nearest-neighbor matching, radius matching, and kernel matching, and the results were robust.

### 4.4. Panel Tobit Regression of Public and Private Hospitals’ Technical Efficiency

In this section, the panel Tobit regression model was used to explore the effects of hospital external characteristics and internal management factors on the technical efficiency of public and private hospitals during 2012–2017. Table 5 presented the regression results of public and private hospitals, respectively.

Considering the hospital characteristics, tertiary hospitals’ technical efficiency was significantly lower than the secondary’s (*p* = 0.009) among public hospitals, but no significant difference was observed between tertiary and secondary private hospitals (*p* = 0.288). Specialized public hospitals were more efficient than general hospitals (*p* < 0.001), while no significant difference in technical efficiency was found among private hospitals with different service types. The governance body of public hospitals had a significant influence on technical efficiency because district (*p* < 0.001) and enterprise or institution (*p* < 0.001) hospitals performed less efficiently than commission hospitals. There was no significant difference in the technical efficiency between for-profit and non-profit private hospitals. Regarding the geographical location, private hospitals located in urban areas had better performance in technical efficiency than those in the suburbs (*p* = 0.005).

Among the internal management factors, bed occupancy rate and annual visits per doctor were positively associated with the technical efficiency of both public and private hospitals (*p* < 0.001), while average length of stay was negatively related to public and private hospitals’ technical efficiency (*p* < 0.001), and ratio of doctors to nurses was only positively related to public hospitals’ technical efficiency (*p* < 0.001). For two variables concerning health expenditures, higher average outpatient cost contributed to lower technical efficiency of public hospitals (*p* < 0.001), and inpatient cost per capita also contributed to lower technical efficiency of public hospitals (*p* = 0.003). As for private hospitals, the average outpatient cost was not significantly associated with the technical efficiency, and higher inpatient cost per capita could lead to higher technical efficiency (*p* = 0.098).

## 5. Discussion

Some interesting and possibly counterintuitive findings are reported in this study. The results of this study presented the essential difference in technical efficiency between public and private hospitals in Beijing in 2012–2017. By comparison, the technical efficiency of private hospitals is much lower than that of public hospitals. In specific, it is found that public hospitals also performed better in pure technical efficiency and scale efficiency. Thus in general, compared with private hospitals, public hospitals can get more output with the same input, and operate with a more suitable scale. Additionally, the majority of private hospitals were in a state of increasing returns to scale in 2012–2017, suggesting that private hospitals should expand their operating scale. McKay [55] found that in the context of payment by Medicare, for-profit private hospitals had a significantly smaller improvement in technical efficiency than government-owned public hospitals in the United States. In addition, many other foreign studies also showed that public hospitals could make better use of inputs so that they provided more effective medical services [19,56], which were consistent with the present study in Beijing, China. However, a study examined the hospital efficiency before and after the extensive hospital privatization beginning in the mid-1990s in Germany, finding that the transformation from public to private for-profit status was associated with an efficiency increase ranging from 2.9% to 4.9% [26], while this study had a premise that only hospital ownership had changed, and other factors such as hospital scale and environment kept the same. In contrast, China is faced with huge gaps between public and private hospitals in various aspects compared to western countries, making it more complex in the analysis of hospital efficiency.

In the comparison of technical efficiency among hospitals with different types of ownership, it is of crucial importance to disentangle the impacts of ownership from the effects of hospital characteristics, patient heterogeneity, market competition, and other confounding factors [19,27]. As China’s healthcare reform deepens, the number of private hospitals is growing rapidly, but their operating scale was still lagging behind public hospitals. Therefore, we adopted PSM analysis to make essential comparisons between the efficiency of private and public hospitals from 2012 to 2017. After matching the propensity scores, the scale efficiency of public hospitals in Beijing was still higher than that of private hospitals, while the pure technical efficiency of public hospitals was surpassed by that of private hospitals, indicating that after removing the influence of other factors, private hospitals have more outputs under the same inputs. Similarly, a study on the efficiency of American urban hospitals found that private hospitals performed more efficiently than public hospitals after controlling hospital characteristics, market characteristics, and patient conditions [26]. According to standard economic theory, private hospitals are predicted to outperform public hospitals in efficiency, and this study can serve as supporting empirical evidence for these theories in the context of modern China. In addition, evidence in the management literature demonstrated that private hospitals were more efficient in management and that their competition with public hospitals could improve the overall efficiency of the healthcare system [31,32], which supported the present study. However, the scale efficiency of private hospitals was still lower than that of public hospitals, suggesting that the lower technical efficiency of private hospitals mainly lies in the lower scale efficiency. This study showed that only 19 private hospitals were in service in 2012, and many private hospitals were still in the ascendant. The Chinese government also reported private hospitals as with “a large number, but small scale” [57], so special attention can be paid to the expansion of private hospitals’ operating scale.

With the participation of private hospitals in the health market, there was no efficiency growth trend in both public and private hospitals in a competitive environment. Therefore, we empirically tested the long-term impacts of hospital characteristics and internal management factors using the panel Tobit regression. The factors affecting the technical efficiency of public and private hospitals were not always the same. For public hospitals, technical efficiency was influenced by various hospital characteristics and internal indicators concerning productivity, health workforce structure, and health expenditures, while for private hospitals, only location and productivity indicators were the main influencing factors. The present study found that urban private hospitals performed better in technical efficiency than those located in the suburb, but other studies did not reach a consistent conclusion on this issue. A recent Canadian study found that the efficiency of rural community hospitals was higher than that of urban community hospitals [58]. On the contrary, Athanassopoulos et al. [59] compared the efficiency of Greek urban and rural hospitals, and concluded that the efficiency of urban hospitals was higher than that of rural hospitals. Under such circumstances, we suggest more attention be paid to the technical efficiency of suburban private hospitals. The governance body of public hospitals is an important issue, and we found that commission hospitals were more efficient than district hospitals and hospitals governed by enterprises or institutions. Burgess and Wilson [20] suggested that federal government hospitals were more efficient than non-federal government hospitals in the United States. Other studies also proposed that there was a large gap among the technical efficiency of federal hospitals, Veterans Administration Affiliated hospitals, and religious non-profit hospitals [60,61], indicating the impact of different governance bodies or management systems on hospital efficiency. Therefore, the Chinese government should make more efforts in improving the efficiency of non-ministerial public hospitals. Many studies concerned about the efficiency of non-profit and for-profit private hospitals. Ozcan et al. [19], as well as Daidone and Amico [62], drew a consistent conclusion that non-profit private hospitals were more efficient than the for-profit ones in the United States and Italy, while Burgess and Wilson [20] indicated an opposite result. The present study, however, found that profitability did not affect the efficiency of private hospitals in Beijing, China.

The present study also showed that both bed occupancy rate and average lengths of stay had impacts on public and private hospitals’ efficiency. Therefore, private hospitals should adopt the necessary measures to reduce the average length of stay and improve bed rotation rates. Annual visits per doctor had a positive effect on both public and private hospitals, indicating that increasing the workload of doctors would improve the technical efficiency of hospitals, but it is of importance to balance the service quality and the interests of health technicians in this process. The ratio of doctors to nurses had a positive sign on the technical efficiency of public hospitals, but another study on county-level general public hospitals in China found that a higher ratio of doctors to nurses is related to lower technical efficiency of public hospitals [44]. A possible explanation for this finding was that the number of annual visits per physician in public hospitals in Beijing was much higher than in private hospitals. Moreover, average outpatient cost and inpatient cost per capita had a negative effect on technical efficiency for public hospitals, so a reduction in health expenditure was recommended to relieve the economic burden on patients and increase the efficiency of public hospitals. In this sense, private hospitals should further expand their scale to improve scale efficiency, while public hospitals should change the hospital’s internal management standards and model, promote refined management, and achieve sustainable development.

The limitations of the present study are as follows. Firstly, only hospitals in Beijing were analyzed in this study, so the results are not nationally representative. Secondly, the reputation of public and private hospitals is fairly different. Public hospitals have better health workforce and equipment, so most public hospitals have a better reputation than private hospitals, and reputation also influences the performance and efficiency of hospitals. However, the present study could not measure the difference in reputation between public and private hospitals. Thirdly, as a commonly shared problem encountered by most developing countries [63], the absence of data regarding case-mix and service quality limits the application value of this study, which can be improved by including health care quality as a variable if required data are available in the future.

## 6. Conclusions

Since the initial 2009 healthcare reform, the Chinese government has been encouraging social capital to sponsor the healthcare market and supporting the development of private hospitals to improve the technical efficiency of hospitals. First, we found that the technical efficiency, pure technical efficiency, and scale efficiency of public hospitals were higher than those of private hospitals. Second, the results by PSM to match “post-randomization” showed that the matched pure technical efficiency of public hospitals became lower than that of private hospitals, while the matched scale efficiency of public hospitals remained higher. Therefore, the ownership of hospitals could affect the hospital’s pure technical efficiency, indicating that private hospitals had better management standards and incorporate scale. Third, with the participation of private hospitals in the health market, there was no efficiency growth trend in both public and private hospitals in a competitive environment. The influencing factors for public hospitals and private hospitals were different. For public hospitals, the current management model can be properly adjusted to improve their management standards, including the reasonable structure of doctors and nurses, appropriate reduction of hospitalization expenses, as well as increasing bed occupancy rate and annual visits per physician. For private hospitals, operating scales should be expanded via proper restructuring, mergers and acquisitions, and they should also pay special attention to shortening the average length of stay and increasing the bed occupancy rate.

## Figures and Tables

**Table 1 ijerph-17-00082-t001:** Statistical description of inputs and outputs variables in public and private hospitals in Beijing in 2012–2017.

Period			2012	2013	2014	2015	2016	2017
Number of hospitals		Public	135	135	138	139	142	145
		private	19	29	46	62	73	87
Inputs	The number of beds	Public	529 ± 438	543 ± 448	561 ± 458	565 ± 464	579 ± 476	586 ± 476
		Private	162 ± 172	134 ± 140	144 ± 156	135 ± 151	126 ± 148	124 ± 161
	Total health technicians	Public	854 ± 777	884 ± 821	908 ± 811	941 ± 847	947 ± 827	960 ± 871
		Private	247 ± 259	207 ± 199	203 ± 212	203 ± 241	182 ± 205	179 ± 217
Outputs	Outpatient and emergency visits	Public	798,551 ± 777,909	861,717 ± 827,150	905,124 ± 874,801	914,980 ± 883,318	955,853 ± 918,608	861,042 ± 848,044
		Private	138,163 ± 236,563	119,322 ± 222,046	92,437 ± 199,095	78,037 ± 180,032	74,168 ± 149,271	70,339 ± 146,690
	Inpatient discharges	Public	14,209 ± 16,763	15,618 ± 18,903	16,690 ± 20,472	17,115 ± 21,253	18,645 ± 23,096	19,310 ± 24,731
		Private	2614 ± 4011	2171 ± 3161	2169 ± 3227	1668 ± 2805	2062 ± 3188	2099 ± 3937
	Revenue (ten thousand Yuan)	Public	57,271 ± 71,348	46,074 ± 56,852	64,651 ± 85,147	63,784 ± 79,201	65,168 ± 80,060	63,739 ± 80,581
		Private	12,580 ± 17,843	10,282 ± 15,113	8889 ± 13,689	7230 ± 12,868	7851 ± 11,820	8095 ± 13,714

Data were shown as mean ± standard deviation (SD).

**Table 2 ijerph-17-00082-t002:** Efficiency of public and private hospitals in Beijing in 2012–2017.

Period		2012	2013	2014	2015	2016	2017
TE	Public	0.589	0.549	0.488	0.503	0.500	0.473
	Private	0.452	0.432	0.362	0.294	0.358	0.315
PTE	Public	0.621	0.576	0.535	0.549	0.529	0.518
	Private	0.604	0.606	0.473	0.401	0.383	0.376
SE	Public	0.952	0.950	0.925	0.925	0.953	0.925
	Private	0.782	0.774	0.825	0.792	0.924	0.841

TE, technical efficiency; PTE, pure technical efficiency; SE, scale efficiency.

**Table 3 ijerph-17-00082-t003:** The number of public and private hospitals in different return to scale states in Beijing in 2012–2017.

Return to Scale States	2012		2013		2014		2015		2016		2017	
	Public	Private	Public	Private	Public	Private	Public	Private	Public	Private	Public	Private
CRS	10	2	9	2	6	3	7	3	8	3	7	3
DRS	44	1	65	1	86	7	86	9	74	3	82	4
IRS	81	16	61	26	44	36	46	50	60	67	56	80
Total	135	19	135	29	138	46	139	62	142	73	145	87

CRS, constant returns to scale; DRS, decreasing returns to scale; IRS, increasing returns to scale.

**Table 4 ijerph-17-00082-t004:** Propensity score matching (PSM) results of private and public hospitals’ PTE and SE in 2012–2017.

Period		2012	2013	2014	2015	2016	2017
TE	Public	0.533	0.466	0.431	0.396	0.354	0.316
	Private	0.513	0.500	0.422	0.374	0.404	0.341
	ATT	0.02	−0.034	0.009	0.022	−0.05	−0.025
PTE	Public	0.556	0.493	0.451	0.414	0.362	0.334
	Private	0.568	0.524	0.458	0.442	0.417	0.362
	ATT	−0.012	−0.031	−0.007	−0.028	−0.055	−0.028
SE	Public	0.960	0.946	0.959	0.957	0.978	0.937
	Private	0.895	0.945	0.922	0.883	0.942	0.923
	ATT	0.065	0.007	0.036	0.074	0.036	0.014

TE, technical efficiency; PTE, pure technical efficiency; SE, scale efficiency; ATT, Average treatment on treated, means the average difference of efficiency score on private hospitals.

**Table 5 ijerph-17-00082-t005:** Panel Tobit regression results of public and private hospitals’ TE in Beijing.

Variables	Public Hospitals			Private Hospitals		
	Coefficient	SE	*p*	Coefficient	SE	*p*
Suburb	ref	ref	ref	ref	ref	ref
Urban	0.029	0.030	0.260	0.116 ***	0.036	0.001
Secondary	ref	ref	ref	ref	ref	ref
Tertiary	−0.036 ***	0.013	0.009	−0.38	0.036	0.288
General	ref	ref	ref	ref	ref	ref
Specialized	0.098 ***	0.028	0.000	−0.024	0.054	0.659
Traditional Chinese medicine	−0.006	0.019	0.744	−0.056	0.058	0.331
Ministerial	ref	ref	ref	-	-	-
Municipal	−0.064	0.041	0.113	-	-	-
District	−0.251 ***	0.038	0.000	-	-	-
Enterprise or Institution	−0.212 ***	0.042	0.000	-	-	-
Non-profit	-	-	-	ref	ref	ref
For-profit	-	-	-	0.018	0.044	0.682
Bed occupancy rate	0.002 ***	2.282 × 10^−4^	0.000	0.003 ***	3.634 × 10^−4^	0.000
Average length of stay	−0.004 ***	6.037 × 10^−4^	0.000	−0.003 ***	5.777 × 10^−4^	0.000
Annual visits per physician	5.120 × 10^−5^ ***	5.590 × 10^−6^	0.000	1.289 × 10^−4^ ***	1.320 × 10^−5^	0.000
Ratio of physicians to nurses	0.064 ***	0.018	0.000	0.010	0.023	0.650
Average outpatient cost	−1.298 × 10^−4^ ***	3.40 × 10^−5^	0.000	3.100 × 10^−5^	2.380 × 10^−5^	0.193
Inpatient cost per capita	−1.240 × 10^−6^ ***	4.230 × 10^−7^	0.003	1.330 × 10^−6^ *	8.060 × 10^−7^	0.098
Constant	0.493 ***	0.0530	0.000	0.005	0.068	0.937

A negative coefficient indicated a negative association with TE and a positive coefficient meant a positive association with TE. *** Significant at the 0.01 level, two-tailed test. * Significant at the 0.10 level, two-tailed test.

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
