# Peer review of "Technical Efficiency of Public and Private Hospitals in Beijing, China: A Comparative Study"

_ijerph, 2019, doi:10.3390/ijerph17010082_

Round 1
Reviewer 1 Report
The proposed article is an application-oriented scientific paper, which sets out to evaluate “the technical efficiency of public and private hospitals in Beijing” and to inform about the “determinants of hospital’s technical efficiency”. To do so, "panel" data on Beijing hospitals is used to calculate technical efficiency, pure technical efficiency and scale efficiency in an output oriented DEA-model.
The DEA-method is well established. Analyses are galore, also in the health sector. While providing several references, this abundance seems not fully reflected in the paper. While this abundance of studies can be seen as an asset concerning viability of the approach, it clearly limits the novelty and the theoretical contribution of the paper. This is fine, as the aim is explicitly to provide policy implications to improve the efficiency of public and private hospitals only.
General appraisal
While the paper is interesting as such and provides at first sight a flawless application of the chosen methods, several aspects (especially concerning the background of the case, theoretical framework, critical appraisal of the methods chosen and the conclusions drawn) need amendment or clarification. Remark: Where possible, line numbers are provided in the review.
The case
Depending on the scope and audience envisaged by IJERPH and the authors, much more information on the background of the provided case is needed for readers not familiar with the Beijing/China health system:
First: How is hospital care funded and governed? Does the governance of hospitals differs between sectors? Did the mode of governance change within the last 10 years, possibly affecting efficiency? Do clients have a free choice across all hospitals? On which basis do they choose? Are health services a luxury good in China? Do health insurance systems alter the choice? Which reputation have private vs. public hospitals? Are hospitals rewarded on a flat rate per case or on daily rates for (occupied) beds? Understanding the system and the incentives given is necessary to make clear that comparing public and private sector providers is feasible. And if so, it would contribute tremendously to the understanding of the differences found.
Second: Please legitimize case selection in more detail: As the authors report, Beijing has a special function in medical service provision in China. Moreover, I would assume that individual income level, competition between hospitals and governance might also differ from other parts of the country. Hence, the sample might be well suited for the comparison envisaged, but by no means “representative” for the country as a whole, or beyond.
Third: What is the effect of the tremendous increase in the private hospital sector? Most private hospitals presumably will have been in business for much less than five years and entered service WITHIN the sampling period (in 2012, only 19 private hospitals were in service!). As such, I would not speak of a panel dataset, as full panel data are available only for few of the private hospitals. Many might be still in the process of setting up operations and develop a customer base. As years are needed to get staff hired, teams formed, processes developed and finally to get a reputation developed among clients and to get beds occupied, it might be too early for an overall system comparison as envisaged in the paper (312: “…operating scale and service quality are still very lagging behind…”). How does the analysis deal with this “start-up” effect, which could bias (not only) scale efficiency? The answer to these questions cannot be propensity score matching only, as to my interpretation, the assumption of selection on observable covariates is not valid in the case at hand: The choice of customers (marginal private benefit and hence client numbers and occupancy rates etc.) will be based on unobserved variables such as quality and cost aspects, signaled by reputation.
Minor remarks:
52: How do you define “progress”?
56f.: How is the “proportion of private hospitals” measured (number, beds, resources?). Did the tremendous growth of the private sector result in a decline in number/beds/resources of public hospitals, or did the total demand for health services just grew exponentially?
Theoretical framework
As mentioned above, the proposed paper is purely analytical. The proposed DEA model linking inputs and outputs is somewhat standard, but not discussed prior analysis. Literature on hospital efficiency drawing on similar models is cited, but the reported contradictory findings in these analyses are not discussed. I would suggest discussing the proposed production model in more detail. The selection of inputs and outputs as well as the internal management factors might need some argumentative backing.
Minor point: why are “revenues” an output? How are “revenues” defined in this respect (157f.)? Is this the total amount of generated funds (inputs), or profits after costs? If the latter, how meaningful are profits as an indicator for public hospitals?
Among the internal management factors (190), I would consider bed occupancy rate, average outpatient cost, and inpatient cost per capita as strongly interlinked and highly influenced by the “start-up” situation described above. While in a conventional analysis I would considered the latter two factors more relevant “outputs” than the output factors defined in the manuscript, the number of produced “cases” can well be considered the outcome of the process. However, as we know nothing about the “quality” of the produced services, this potential lack of comparability should be discussed and conclusions might be formulated more cautious.
Please check your understanding of the function of theory (66f.): “The advantage of private for-profit ownership over public ownership has been proven by different economic theories…”
Methods
Please excuse when I missed some aspects in the paper – if not done already, could you detail the following points?
Please improve the descriptions of the definitions of all factors included into the equation. Could you provide more information on the homogeneity of the units under assessment (SDs are huge, maybe scatterplots of the main hospital characteristics help to understand the state of the field). Possibly, a more critical discussion of the approaches chosen and the case and data might be helpful (see Dyson et al. 2001: Pitfalls and protocols in DEA). Propensity score matching alters the empirical results significantly. Therefore, special attention has to be given to the methodological aspects of this step for making the results (more) convincing. On which factors the propensity score matching is based? Is the approach viable when only few (as in 2012: 19) cases can be found in one of the categories and most of these cases most possibly cluster on one end of the spectrum of hospital characteristics (e.g. small, specialized and/or first level private hospitals?). Please report the balance of baseline characteristics between public and private hospitals in the matched sample. Do you control for “patient heterogeneity, market competition and other confounding factors (310)? Why do you not complement your analyses with a straightforward regression across both systems, inserting private and public as dummy variable?
Results
Several of the reported conclusions seem either trivial or are not fully backed by the findings. Possibly to the DEA-focus some interesting “real world” observations are ignored as the state of the private field and instructive descriptive statistics reported before (e.g. the number of annual visits per physician are MUCH higher in total numbers in public than in private hospitals - this could have several reasons worthwhile to discuss (and could explain the higher physicians/nurses ratio!). I would therefore suggest taking a more cautious approach in the interpretation of the results.
Could you add some more sense to or skip this phrase: “Therefore, we propose that public hospitals should balance the proportion of doctors, nurses to ensure long-run efficiency.” (360).
Author Response
The case
Depending on the scope and audience envisaged by IJERPH and the authors, much more information on the background of the provided case is needed for readers not familiar with the Beijing/China health system:
First: 1) How is hospital care funded and governed? Does the governance of hospitals differ between sectors?
Response: There were two types of hospital ownership in China: public and private. The public hospitals are funded by governments, such as National Health Commission of China, provincial, municipal, county or district governments, and public enterprises or institutions. The private hospitals are funded by private enterprises or individuals. Private hospitals can implement the same co-payment rates as public hospitals if they have signed contracts with health insurance programs. Therefore, both public and private hospitals are paid by health insurance and patients’ out-of-pocket money based on a fee-for-service payment mechanism. However, on average most public hospitals have to earn 90% of their revenues from services provided, with government direct subsidies making up the rest.
2) Did the mode of governance change within the last 10 years, possibly affecting efficiency?
The mode of governance did not change much within the last 10 years, which should not affect efficiency.
3) Do clients have a free choice across all hospitals? On which basis do they choose?
Clients or patients can freely choose from all hospitals, as public and private hospitals adopt the same co-payment rates. They may choose primary care clinics or hospitals with different co-payment rates.
4) Are health services a luxury good in China?
Based on economic theory, health service is a luxury good. Because of the rapid aging and economic development, health service has become increasingly more important to people and thus is regarded as a luxury good.
5) Do health insurance systems alter the choice?
Because private hospitals charge the same co-payment rates under the same health insurance, health insurance won’t alter the choice of public and private hospitals.
6) Which reputation has private vs. public hospitals?
The reputation of public and private hospitals is fairly different. Public hospitals have better health workforce and equipment, so most public hospitals have a better reputation than private hospitals.
.
7) Are hospitals rewarded on a flat rate per case or on daily rates for (occupied) beds?
Hospitals are usually paid by fee-for-service based on what services are provided to patients including the daily rates for occupied beds. A few hospitals are rewarded on a flat rate per case.
8) Understanding the system and the incentives given is necessary to make clear that comparing public and private sector providers is feasible. And if so, it would contribute tremendously to the understanding of the differences found.
We have added much more information about the health care system and incentives in China in the introduction section.
Second: Please legitimize case selection in more detail: As the authors report, Beijing has a special function in medical service provision in China. Moreover, I would assume that individual income level, competition between hospitals and governance might also differ from other parts of the country. Hence, the sample might be well suited for the comparison envisaged, but by no means “representative” for the country as a whole, or beyond.
Response: Beijing has more medical resources in both public and private health sectors, and consequently, hospitals in Beijing not only provide medical services for local residents, but also attract patients across the entire country. The number and quality of public and private hospitals in Beijing may be different from other places, so we agree with the reviewer that our sample might be well suited for the comparison envisaged, but by no means “representative” for the country as a whole, or beyond. In addition, since the announcement of promoting hospitals with social capitals in China, the private hospitals have sprung up in all urban areas. Therefore, this study chose Beijing as a representative trend to compare the efficiency of public and private hospitals, which can provide some information and inspiration to other urban parts of China, and even contribute to the development of public and private hospitals in other developing countries. Please see page 3 in our revised manuscript (clean version), in which we further explained this question.
Third: What is the effect of the tremendous increase in the private hospital sector? Most private hospitals presumably will have been in business for much less than five years and entered service WITHIN the sampling period (in 2012, only 19 private hospitals were in service!). As such, I would not speak of a panel dataset, as full panel data are available only for few of the private hospitals. Many might be still in the process of setting up operations and develop a customer base. As years are needed to get staff hired, teams formed, processes developed and finally to get a reputation developed among clients and to get beds occupied, it might be too early for an overall system comparison as envisaged in the paper (312: “…operating scale and service quality are still very lagging behind…”). How does the analysis deal with this “start-up” effect, which could bias (not only) scale efficiency? The answer to these questions cannot be propensity score matching only, as to my interpretation, the assumption of selection on observable covariates is not valid in the case at hand: The choice of customers (marginal private benefit and hence client numbers and occupancy rates etc.) will be based on unobserved variables such as quality and cost aspects, signaled by reputation.
Response: We have used the unbalanced panel data set, and also agreed with the reviewer’s comments. Private hospitals might be still in the process of setting up operations and are developing a customer base, but most private hospitals in our sample actually existed before 2012 (the first year we studied). Some hospitals were upgraded from primary care clinics to hospitals, which were added into the data sets (primary care clinics were not surveyed). The present study examined the difference of average technical efficiency, pure technical efficiency and scale efficiency between public and private hospitals (excluding primary care clinics) from 2012 to 2017. We are not able to distinguish the start-up effects from the ownership, since some private hospitals entered the market after 2012 but all the public hospitals exited before 2012.
Minor remarks:
52: How do you define “progress”?
Response: The “progress” that appeared in the first paragraph of introduction section referred to “the development of technology”.
56f.: How is the “proportion of private hospitals” measured (number, beds, resources?). Did the tremendous growth of the private sector result in a decline in number/beds/resources of public hospitals, or did the total demand for health services just grew exponentially?
Response: The “proportion of private hospitals” on page 2 is in terms of hospital numbers and the market shares of inpatient and outpatient services. From 2009 to 2017 in China, the number of private hospitals increased from 6240 (30.75% of the total number of hospitals including both public and private hospitals) to 18759 (60.40%), the market share for outpatient services in the private sector grew form 8% to 14.2, and the market share for inpatient services in the private sector increased from 8% to 17.6%.
In addition, since the Chinese government encouraged the development of private hospitals in 2010, some extra constraints for public hospitals have been taken such as the banning of new building projects or expansion of beds.
Theoretical framework
As mentioned above, the proposed paper is purely analytical. The proposed DEA model linking inputs and outputs is somewhat standard, but not discussed prior analysis. Literature on hospital efficiency drawing on similar models is cited, but the reported contradictory findings in these analyses are not discussed. I would suggest discussing the proposed production model in more detail. The selection of inputs and outputs as well as the internal management factors might need some argumentative backing.
Response: We have added a literature review section in which contradictory findings are reported. Please see pages 3-4 in the revised manuscript (clean version). With the participation of private hospitals in the health care market, there was more competition between public and private hospitals. Private hospitals would expand their sizes, increase the number of beds, and attract more health technicians. Therefore, this study used the number of beds and the number of technicians as input indicators. For hospitals in China, the most important health output indicators are outpatient and emergency visits, and inpatient discharges.
In addition, we discussed the proposed production model in details, and added more argumentative backing about the selection of inputs and outputs as well as the internal management factors. Please see line 200-239 in revised manuscript (clean version).
Minor point: why are “revenues” an output? How are “revenues” defined in this respect (157f.)? Is this the total amount of generated funds (inputs), or profits after costs? If the latter, how meaningful are profits as an indicator for public hospitals?
Response: The “revenue” was the total amount of medical incomes for hospitals, instead of profits after deducting costs. We used revenues as an output measure for the following reasons: 1) most public hospitals have to earn 90% of their revenues from services provided, with government direct subsidies making up the rest, which is popular across China. In this way, many public hospitals act as private entities, putting profits above patient welfare [He et al., 2011; Ou et al., 2014]. Medical staff are the residual claimants of profits in public hospitals, and they are the actual shareholders of public hospitals. The revenues/medical incomes also act as the main output indicator for private hospitals. 2) A review in China about choices of input and output indicators in DEA model used total revenue as an output [Wang et al., 2016].
Among the internal management factors (190), I would consider bed occupancy rate, average outpatient cost, and inpatient cost per capita as strongly interlinked and highly influenced by the “start-up” situation described above. While in a conventional analysis I would considered the latter two factors more relevant “outputs” than the output factors defined in the manuscript, the number of produced “cases” can well be considered the outcome of the process. However, as we know nothing about the “quality” of the produced services, this potential lack of comparability should be discussed and conclusions might be formulated more cautious.
Response: This study divided influencing factors into two categories: hospital characteristics and internal management factors. Private hospitals might be still in the process of setting up operations and are developing a customer base, but most private hospitals in our sample actually existed before 2012 (the first year we studied). Some hospitals were upgraded from primary care clinics to hospitals, which were added into the data sets (primary care clinics were not surveyed). We admitted that upgrading from primary care clinics to hospitals may still have some start-up effects, but we were not able to distinguish them from the effects of ownership. The present study focused on the difference of average technical efficiency, pure technical efficiency and scale efficiency between public and private hospitals (excluding primary care clinics) from 2012 to 2017.
Please check your understanding of the function of theory (66f.): “The advantage of private for-profit ownership over public ownership has been proven by different economic theories…”
Response: We have deleted the discussion about the function of theory.
Methods
Please excuse when I missed some aspects in the paper – if not done already, could you detail the following points?
Please improve the descriptions of the definitions of all factors included into the equation. Could you provide more information on the homogeneity of the units under assessment (SDs are huge, maybe scatterplots of the main hospital characteristics help to understand the state of the field). Possibly, a more critical discussion of the approaches chosen and the case and data might be helpful (see Dyson et al. 2001: Pitfalls and protocols in DEA). Propensity score matching alters the empirical results significantly. Therefore, special attention has to be given to the methodological aspects of this step for making the results (more) convincing. On which factors the propensity score matching is based? Is the approach viable when only few (as in 2012: 19) cases can be found in one of the categories and most of these cases most possibly cluster on one end of the spectrum of hospital characteristics (e.g. small, specialized and/or first level private hospitals?). Please report the balance of baseline characteristics between public and private hospitals in the matched sample. Do you control for “patient heterogeneity, market competition and other confounding factors (310)? Why do you not complement your analyses with a straightforward regression across both systems, inserting private and public as dummy variable?
Response: We have used the propensity score matching (PSM) to balance two different types of hospital ownership. The PSM was based on hospitals’ level, service type, geographical location, the number of beds, and the number of health technicians, revenues, average outpatient cost and inpatient cost per capita. We thought these factors could balance the difference and heterogeneity between public and private hospitals.
In addition, in order to compare the efficiency values with DEA results, we would like to get the value of technical efficiency, pure technical efficiency and scale efficiency after controlling for other confounders, therefore, so we used the PSM methodology. However, the regression across systems, inserting private and public as dummy variable did not enable us to get the value.
Results
Several of the reported conclusions seem either trivial or are not fully backed by the findings. Possibly to the DEA-focus some interesting “real world” observations are ignored as the state of the private field and instructive descriptive statistics reported before (e.g. the number of annual visits per physician are MUCH higher in total numbers in public than in private hospitals - this could have several reasons worthwhile to discuss (and could explain the higher physicians/nurses ratio). I would therefore suggest taking a more cautious approach in the interpretation of the results.
Response: We have taken a more cautious approach in the interpretation of the results, accordingly. The ratio of doctors to nurses had a positive sign on the technical efficiency of public hospitals in Beijing, while another study on county-level general public hospitals in China found that higher ratio of doctors to nurses is related to lower technical efficiency of public hospitals. We have added some discussion according to Reviewer 1’s comment, and a possible explanation was that the number of annual visits per physician in public hospitals in Beijing was much higher than private hospitals. Please see line 409-415 in the revised manuscript (clean version).
Could you add some more sense to or skip this phrase: “Therefore, we propose that public hospitals should balance the proportion of doctors, nurses to ensure long-run efficiency.” (360).
Response: We have skipped the phrase according to Reviewer 1’s comment.
Reviewer 2 Report
Review report:
Technical eciency of public and private hospitals in Beijing, China: a comparative study
Manuscript ID: ijerph-631069
1 Summary
This paper presents an empirical study of the eciency of Chinese hospitals located in Beijing. The study is based on an unbalanced panel of public and private hospitals, observed during the period of 2012-2017. The authors apply Data Envelopment Analysis (DEA), Propensity Score Matching (PSM) and Tobit regression to evaluate dierent eciency measures, including Technical Eciency (TE), Pure Technical Eciency (PTE) and Scale Eciency (SE). Their ndings suggest that all measures were higher for public hospitals relative to private hospitals in Beijing.
2 Comments
The research topic is relevant and worth consideration. The dataset is recent and valuable, covering dierent types of hospitals and including a number of interesting variables related to eciency measurement. However, the paper lacks a proper description of the proposed methods and a clear explanation of how the authors obtained their results from the application of these methods to the data. In my opinion, the distinction between input, output and explanatory variables proposed in Table 1 is questionable and possibly weakens the validity of the empirical results. The main issues are detailed in the following comments.
Introduction: The Introduction provides an extensive background for the paper, including 36 references to studies of health care eciency and related theories. However, many of these references date from the 1970s-1990s and a number of them is redundant (e.g., references 14 to 19 pertain to similar topics and are not very recent). I would suggest to concentrate on the focus of the paper and connect it directly to recent work on health care eciency and DEA, indicating clearly the need for the proposed study and outlining its main activity/ndings.
Material and Methods: In my view, this section should be rewritten to provide a clear understanding of the data and the proposed methods. In particular,
In the Data Section (2.1), the authors report using a panel of 241 hospitals in 2012-2017. However, Table 1 shows that the panel is unbalanced and the number of hospitals varies from 154 in 2012 to 232 in 2017. The panel composition, its compatibility with the proposed methods and its in uence on the results should be discussed in detail.
The purpose of the study is to compare the eciency of public and private hospitals. On p. 3, the hospitals included in the dataset were labeled either as secondary (100-499 beds) or tertiary (over 500 beds). However, the authors use pooled data and do not report the relative frequencies of public and private hospitals that are classied, respectively, as secondary and tertiary. In my opinion, this approach could lead to biased results since the proportion of small scale structures is much higher among private hospitals (p. 3, line 120) and this is a likely explanation for their lower (scale) eciency and increasing returns to scale (Table 3). It is not clear how the authors applied Propensity Score Matching (Section 2.3) and whether their technique is able to deal with this size effect.
The choice of inputs and outputs for DEA is based on the leading textbook by Marshall (1890). However, a number of recent papers have considered the problem specically for hospital inputs and outputs, suggesting a number of important variables for consideration. In my opinion, the classication of inputs, outputs and explanatory variables proposed in Table 1 (p. 4) is questionable and possibly misleading. In particular, the authors classify Bed occupancy, Average length of stay and Annual visits per physician as \Explanatory variables", whereas a number of studies (see, e.g., Azreena et al., 2018, Li et al., 2019, Stefko et al., 2018) indicate that these variables should be regarded as hospital outputs. I would suggest to carefully reconsider the de nition of input and output variables for DEA (and Tobit regression) in light of the recent literature.
Please specify the main equations for DEA and PSM, indicating clearly how you apply them to your data.
3 Suggested references
Azreena E., Muhamad H.J., Rosliza A.M. (2018). A systematic review of hospital inputs and outputs in measuring technical effciency using data envelopment analysis. International Journal of Public Health and Clinical Sciences, 5(1), 17-35.
Li B., Mohiuddin M., Liu, Q. (2019). Determinants and dierences of township hospital eciency among Chinese provinces. International Journal of Environmental Research and Public Health, 16(9), 1601, https://doi.org/10.3390/ijerph16091601.
Stefko R., Gavurova B., Kocisova K. (2018). Healthcare effciency assessment using DEA analysis in the Slovak Republic, Health Economics Review, https://doi.org/10.1186/s13561-018-0191-9.
Author Response
Introduction: The Introduction provides an extensive background for the paper, including 36 references to studies of health care efficiency and related theories. However, many of these references date from the 1970s-1990s and a number of them is redundant (e.g., references 14 to 19 pertain to similar topics and are not very recent). I would suggest concentrating on the focus of the paper and connecting it directly to recent work on health care efficiency and DEA, indicating clearly the need for the proposed study and outlining its main activity/findings.
Response: We thank Reviewer 2 for helpful comments on We have deleted the references published early in the 1970s-1990s, and added a new section of the literature review into the revised manuscript. It directly focused on the recent work on health care efficiency and DEA, indicating clearly the need for the proposed study and outlining its main activities/findings.
Material and Methods: In my view, this section should be rewritten to provide a clear understanding of the data and the proposed methods. In particular, In the Data Section (2.1), the authors report using a panel of 241 hospitals in 2012-2017. However, Table 1 shows that the panel is unbalanced and the number of hospitals varies from 154 in 2012 to 232 in 2017. The panel composition, its compatibility with the proposed methods and its influence on the results should be discussed in detail.
Response: The panel composition, its compatibility with the proposed methods and its influence on the results in the maternal and methods have been discussed in detail. Please see line 146-246 in the revised manuscript (clean version).
The purpose of the study is to compare the efficiency of public and private hospitals. On p. 3, the hospitals included in the dataset were labeled either as secondary (100-499 beds) or tertiary (over 500 beds). However, the authors use pooled data and do not report the relative frequencies of public and private hospitals that are classified, respectively, as secondary and tertiary. In my opinion, this approach could lead to biased results since the proportion of small scale structures is much higher among private hospitals (p. 3, line 120) and this is a likely explanation for their lower (scale) efficiency and increasing returns to scale (Table 3). It is not clear how the authors applied Propensity Score Matching (Section 2.3) and whether their technique is able to deal with this size effect.
Response:As Reviewer 2 suggested, we ran analyses again and reported the technical efficiency, pure technical efficiency and scale efficiency of public and private hospitals that were classified, respectively, as secondary and tertiary. In terms of different hospital levels, both pure technical and scale efficiency of tertiary public hospitals were higher than those of private hospitals, but for secondary public hospitals, the pure technical efficiency was lower than private hospitals, while the scale efficiency was better than that of private hospitals. Please see appendix 1.1-1.2, which supported our conclusions.
Appendix 1.1. Efficiency of public and private tertiary hospitals in Beijing in 2012-2017.
|
Period |
2012 |
2013 |
2014 |
2015 |
2016 |
2017 |
|
|
TE |
Public |
0.706 |
0.675 |
0.582 |
0.601 |
0.603 |
0.567 |
|
Private |
0.396 |
0.340 |
0.382 |
0.352 |
0.359 |
0.308 |
|
|
PTE |
Public |
0.728 |
0.705 |
0.656 |
0.663 |
0.648 |
0.628 |
|
Private |
0.399 |
0.358 |
0.389 |
0.380 |
0.379 |
0.331 |
|
|
SE |
Public |
0.973 |
0.961 |
0.902 |
0.918 |
0.940 |
0.914 |
|
Private |
0.996 |
0.880 |
0.968 |
0.912 |
0.947 |
0.929 |
|
Appendix 1.1. Efficiency of public and private secondary hospitals in Beijing in 2012-2017.
|
Period |
2012 |
2013 |
2014 |
2015 |
2016 |
2017 |
|
|
TE |
Public |
0.509 |
0.446 |
0.400 |
0.391 |
0.381 |
0.361 |
|
Private |
0.459 |
0.443 |
0.360 |
0.282 |
0.357 |
0.316 |
|
|
PTE |
Public |
0.547 |
0.469 |
0.422 |
0.418 |
0.392 |
0.387 |
|
Private |
0.629 |
0.635 |
0.483 |
0.405 |
0.384 |
0.387 |
|
|
SE |
Public |
0.937 |
0.940 |
0.947 |
0.933 |
0.967 |
0.937 |
|
Private |
0.757 |
0.762 |
0.808 |
0.766 |
0.918 |
0.820 |
|
The choice of inputs and outputs for DEA is based on the leading textbook by Marshall (1890). However, a number of recent papers have considered the problem especially for hospital inputs and outputs, suggesting a number of important variables for consideration. In my opinion, the classification of inputs, outputs and explanatory variables proposed in Table 1 (p. 4) is questionable and possibly misleading. In particular, the authors classify Bed occupancy, Average length of stay and Annual visits per physician as \Explanatory variables", whereas a number of studies (see, e.g., Azreena et al., 2018, Li et al., 2019, Stefko et al., 2018) indicate that these variables should be regarded as hospital outputs. I would suggest carefully reconsidering the definition of input and output variables for DEA (and Tobit regression) in light of the recent literature.
Response: The reasons for choosing these input and output indicators were shown in the revised manuscript. We carefully studied the references recommended by Reviewer 2, and benefited from them a lot. We choose these indicators based on the textbook by Marshall (1890) and many new references, and finally took the number of total health technicians including the number of doctors, nurses, pharmacists, laboratorians and other staff, and the number of beds as input indicators. According to a study suggested by reviewer [Azreena et al., 2018], most studies would include the number of beds and the number of staffs (different choice) as inputs. Please see line 194-199 in our revised manuscript (clean version).
The number of outpatient visits and the number of discharged patients were the indicators often used in the previous studies, and most public hospitals have to earn 90% of their revenue from services provided, with government direct subsidies making up the rest, which is common across China. In this way, many public hospitals act as private entities, putting profit above patients’ welfare [He et al., 2011; Ou et al., 2014]. Physicians are the residual claimants of profits in public hospitals, and they are the actual shareholders of the public hospitals. The revenue/medical income also acts as the main output indicator for private hospitals. In addition, A review in China about the choice of input and output indicators in DEA indicated the total revenue as an output indicator [Wang et al., 2016], which is consistent with our study. Therefore, the revenue acted as an output indicator for Chinese public and private hospitals in the present study. Please see line 201-205 in our revised manuscript (clean version).
As for the bed occupancy rate, average length of stay and annual visits per physician, we also chose them as explanatory variables based on a large number of references, in which these factors were usually considered as determinants. We thought these factors were not the main outputs for this study. Please see line 233-239 in our revised manuscript (clean version).
Please specify the main equations for DEA and PSM, indicating clearly how you apply them to your data.
Response: We have specified the main equations for DEA and PSM, and indicated clearly how we applied them to our data.
Please see the line 187-223 in our revised manuscript (clean version).
Suggested references
Azreena E., Muhamad H.J., Rosliza A.M. (2018). A systematic review of hospital inputs and outputs in measuring technical effciency using data envelopment analysis. International Journal of Public Health and Clinical Sciences, 5(1), 17-35.
Li B., Mohiuddin M., Liu, Q. (2019). Determinants and di
erences of township hospital efficiency among Chinese provinces. International Journal of Environmental Research and Public Health, 16(9), 1601, https://doi.org/10.3390/ijerph16091601.
Stefko R., Gavurova B., Kocisova K. (2018). Healthcare effciency assessment using DEA analysis in the Slovak Republic, Health Economics Review, https://doi.org/10.1186/s13561-018-0191-9.
Response: Thank you for these useful references.
Reviewer 3 Report
Dear authors,
Here are the notes to consider:
1) Abstract: from abstract (but also from the whole text) personal phrases should be eliminated, e.g. "we employed" etc. The abstract should have the following structure: Aim, methods, tools, results, discussion.
2) Introduction. In the introduction, the reader should be introduced to the thought process to describe why the problem and methods were used. A full literature review should not be made in the introduction. The purpose should be clarified. In its current form it is not clear enough.
3) Add section: literature review. A critical analysis of the literature should be carried out, which will lead to the identification of research gaps that the authors intend to fill with their research analysis.
4) the methodology should be refined - why these methods were chosen? are there other better or worse? what is the justification? The research hypothesis (or hypotheses) should be clarified. Its absence deprives logic of conducting research and analysis. The hypothesis and results in the discussion should be verified.
5) Future research directions should be specified on the basis of the analysis carried out. Can the test results be transferred to other areas or other countries? What is the level of research universality or what is the specificity for choosing only these hospitals?
In principle, the DEA and PSM analysis was carried out correctly.
Good luck!
Author Response
1) Abstract: from abstract (but also from the whole text) personal phrases should be eliminated, e.g. "we employed" etc. The abstract should have the following structure: Aim, methods, tools, results, discussion.
Response: Many thanks for Reviewer 3’s helpful comments. We have eliminated personal phrases as Reviewer 3 suggested. In addition, the abstract had the following structure: objective (aim), methods, results, and conclusions.
2) Introduction. In the introduction, the reader should be introduced to the thought process to describe why the problem and methods were used. A full literature review should not be made in the introduction. The purpose should be clarified. In its current form it is not clear enough.
Response: We have restructured this section accordingly. In the revised manuscript, we firstly introduced the importance of studies on hospital technical efficiency in China. Secondly, we summarized the Chinese health system and the development of and difference between public and private hospitals since the new health care reform in 2009, and further analyzed the necessity and feasibility of studying efficiency of public and private hospitals in China. Thirdly, we explained the reason why hospitals in Beijing were analyzed. In addition, we have moved the full literature review from the introduction section to the new section of literature review in the revised manuscript.
3) Add section: literature review. A critical analysis of the literature should be carried out, which will lead to the identification of research gaps that the authors intend to fill with their research analysis.
Response: We have added a new section of literature review into the revised manuscript and critically analyzed the previous studies, and we finally identified the research gaps to which our study intend to fill. Please see Pages 3-4 in the revised manuscript (clean version).
4) The methodology should be refined - why these methods were chosen? are there other better or worse? what is the justification? The research hypothesis (or hypotheses) should be clarified. Its absence deprives logic of conducting research and analysis. The hypothesis and results in the discussion should be verified.
Response: We have refined the methodology. DEA allows for simultaneous consideration of multiple inputs and outputs, which is suitable for measuring the efficiency of complex service organizations like hospitals. Under the current healthcare system in China, the two ownership types of hospitals (public and private) vary in scale, level, service type, number of beds, personnel, etc., so it is difficult to distinguish the impact of ownership or other factors. The propensity score matching (PSM), proposed by Rosenbaum and Rubin, can create a “quasi-random” test. Adjusting the scalar propensity score is sufficient to remove bias due to all observed covariates, so we use PSM methods to get value of efficiency after matching between public and private hospitals. Censored efficiency scores (0-1, concentrate on boundary values) cannot be obtained via the ordinary least squares method, so it is preferable to estimate coefficients using the panel Tobit model. For censored data using Tobit model, there is insufficient statistic of individual heterogeneity allowing the fixed effects to be conditioned out of likelihood, so Tobit model with fixed effects is usually not recommended, and the methods can also be applied to the unbalanced data set.
5) Future research directions should be specified on the basis of the analysis carried out. Can the test results be transferred to other areas or other countries? What is the level of research universality or what is the specificity for choosing only these hospitals?
Response: Beijing has more medical resources in both public and private health sectors, and consequently, hospitals in Beijing not only provide medical services for local residents, but also attract patients across the entire country. The number and quality of public and private hospitals in Beijing may be different from other places. In addition, since the announcement of promoting hospitals with social capitals in China, the private hospitals have sprung up in Beijing, and have been able to compete effectively with public hospitals. Therefore, the competition trend between public hospitals and private hospitals is the similar in different regions of China. Please see pages 2-3 in our manuscript (clean version).
In principle, the DEA and PSM analysis was carried out correctly.
Response: Many thanks for Reviewer 3’s favorable comments on our measures of DEA and PSM.
Round 2
Reviewer 1 Report
Thank you very much for the revision and the detailed responses to both reviewers’ comments.
While still maintaining its somewhat atheoretical, purely analytical character, the manuscript gained significantly with the revision. Thank you for providing precise information on case and methods and taking a more cautious approach in the interpretation of your results. A few points might be improved in my view:
The theoretical model is still the weakest part of the paper. While after several reads I can follow your reasoning now, your motivation could be outlined to the reader at the beginning of the paper (abstract, intro) more precisely.
This could be done the easiest by outlining AS WELL what you are NOT doing… hence explaining performance in terms of quality, management, reputation etc. pp. Table 1 is still puzzling, differentiating between “inputs” and “explanatory variables”. This implies, you might outline the reasoning behind the chosen design of your production model in more detail (still). Don’t get me wrong: your model might make sense when elaborating on technical efficiency as defined. However, you might still outline your causal reasoning (maybe by making your implicit hypotheses explicit) and explain while alternative configurations (as mentioned by both reviewers) are dismissed (194ff.). Please make (even more) clear, why revenues qualify as an output and why the explanatory variables do not. I would suggest to provide a simple graph of the causal model you are proposing.
In this context, you might reformulate the following sentences in line 203ff.: “Many public hospitals acted as private entities, putting profit above patient welfare, and medical staff was the residual claimants of profits [48, 49]. Therefore, the revenue is also regarded as an output indicator in present study.”
I find these sentences extremely puzzling, as they not only seem to address so many different spheres, but contain so many contradictions (when not totally immersed in your mode of reasoning) that it might be hard to convince your readers (one could read: public clinics are private; and have shareholders in China? Public clinics endanger patients? Maximizing revenues is a bad thing, indicating this form of high jacking?). Sorting this points out might improve the manuscript significantly.
The conclusion became extremely brief. At least (re) state the specific goal and not-goals of the study and link up more tightly the results with your interpretations.
Please carefully language edit the additions made in the revision.
Author Response
Comments and Suggestions for Authors
Thank you very much for the revision and the detailed responses to both reviewers’ comments.
While still maintaining its somewhat a theoretical, purely analytical character, the manuscript gained significantly with the revision. Thank you for providing precise information on case and methods and taking a more cautious approach in the interpretation of your results. A few points might be improved in my view:
The theoretical model is still the weakest part of the paper. While after several reads I can follow your reasoning now, your motivation could be outlined to the reader at the beginning of the paper (abstract, intro) more precisely.
Response:Many thanks for Reviewer 1’s helpful comments. We outlined motivation at the beginning of paper (abstract, intro) more precisely. Please see the sections of Abstract and Introduction in the revised manuscript.
This could be done the easiest by outlining AS WELL what you are NOT doing… hence explaining performance in terms of quality, management, reputation etc.
Response: We have added what we are not able to explain performance between public and private hospitals in terms of quality, reputation etc. as a limitation. Please see line 573-581 in our revised manuscript.
Table 1 is still puzzling, differentiating between “inputs” and “explanatory variables”. This implies, you might outline the reasoning behind the chosen design of your production model in more detail (still). Don’t get me wrong: your model might make sense when elaborating on technical efficiency as defined. However, you might still outline your causal reasoning (maybe by making your implicit hypotheses explicit) and explain while alternative configurations (as mentioned by both reviewers) are dismissed (194ff.). Please make (even more) clear, why revenues qualify as an output and why the explanatory variables do not. I would suggest to provide a simple graph of the causal model you are proposing.Response: In order to avoid ambiguity, we have transferred the descriptive statistics of explanatory variable from Table1 to the Appendix 1.
Based on the background that an increasing amount of private hospitals are enrolled into Chinese health market, our study intended to compare the technical efficiency, pure technical efficiency and scale efficiency between public and private hospitals in China. In addition, we analyzed the influencing factors of public and private hospitals’ efficiency.
Due to the limitations of data availability, we explained in detail the reasons for choosing our input and output variables. Please see line 279-295 in our revised manuscript.
We used revenues as an output measure for the following reason. Most public hospitals have to earn 90% of their revenues from the medical services they provide, and the direct subsidies from the government make up the rest, which is very popular across China. The revenues also act as the main output indicator for private hospitals. Therefore, we believed that the revenue was the main output indicator in this study, rather than explanatory variable.
We used multiple input and output indicators and weighted them to obtain technical efficiency by using DEA. We assumed that the influencing factors of different technical efficiency mainly included the hospital characteristics and internal management in Beijing. Our hypothesis is that these factors affected hospital technical efficiency, but we admit that it is difficult to ascertain causal inferences in this study.
In this context, you might reformulate the following sentences in line 203ff.: “Many public hospitals acted as private entities, putting profit above patient welfare, and medical staff was the residual claimants of profits [48, 49]. Therefore, the revenue is also regarded as an output indicator in present study.” I find these sentences extremely puzzling, as they not only seem to address so many different spheres, but contain so many contradictions (when not totally immersed in your mode of reasoning) that it might be hard to convince your readers (one could read: public clinics are private; and have shareholders in China? Public clinics endanger patients? Maximizing revenues is a bad thing, indicating this form of high jacking?). Sorting this points out might improve the manuscript significantly.
Response:As a matter of fact, many hospitals in China, although in the name of being public, were in effect operating as private entities, putting profit above patient welfare. However, with the health care system reform in China, public hospitals are gradually driven to pursue people’s welfare. It is not a good practice for public hospitals to maximize revenues, so we strongly agree with Reviewer 1. Therefore, we pointed it out in the revised manuscript. Please see line 290-291, as we add “Although maximizing revenues is inappropriate for public hospitals, many of them take it as an important measure for their performance in China. Therefore, the revenue is also enrolled as an output indicator in the present study.” into our revised manuscript
The conclusion became extremely brief. At least (re) state the specific goal and not-goals of the study and link up more tightly the results with your interpretations.
Response: We restructured the conclusion section and stated the specific goal and not-goals of the study and link up more tightly the results with our interpretations. Main conclusions were as follows.
First, we found that the technical efficiency, pure technical efficiency and scale efficiency of public hospitals were higher than those of private hospitals. Second, the results by PSM to match “post-randomization” showed that the matched pure technical efficiency of public hospitals became lower than that of private hospitals, while the matched scale efficiency of public hospitals remained higher. Therefore, the ownership of hospitals could affect hospital’s pure technical efficiency, indicating that private hospitals had better management standards and incorporate scale. Third, with the participation of private hospitals in the health market, there was no efficiency growth trend in both public and private hospitals in a competitive environment. The influencing factors for public hospitals and private hospitals were different. For public hospitals, current management model can be properly adjusted to improve their management standards, including reasonable structure of doctors and nurses, appropriate reduction of hospitalization expenses, as well as increasing bed occupancy rate and annual visits per physician. For private hospitals, operating scales should be expanded via proper restructuring, mergers and acquisitions, and they should also pay special attention to shortening the average length of stay and increasing the bed occupancy rate.
Please carefully language edit the additions made in the revision.
Response: We have carefully edited the language.
Reviewer 2 Report
The authors have considered all the proposed suggestions and have (partially) implemented them. The quality of the manuscript has improved, with a clear presentation of the methodology and a more detailed description of the dataset.
I would still recommend a careful revision of English language and style.
Author Response
Comments and Suggestions for Authors
The authors have considered all the proposed suggestions and have (partially) implemented them. The quality of the manuscript has improved, with a clear presentation of the methodology and a more detailed description of the dataset.
Response: Many thanks for Reviewer 2’s favorable comments.
I would still recommend a careful revision of English language and style.
Response: We carefully revised the language and style.
Reviewer 3 Report
The authors applied all suggested changes. I have no more comments.
Author Response
Comments and Suggestions for Authors
The authors applied all suggested changes. I have no more comments.
Response: Many thanks for Reviewer 3’s favorable comments.